# Impact of Active Physiotherapy Rehabilitation on Pain and Global and Functional Improvement 1–2 Months after Lumbar Disk Surgery: A Systematic Review and Meta-Analysis

**DOI:** 10.3390/healthcare10101943

**Published:** 2022-10-05

**Authors:** Kiran Afzal, Hafsah Gul Khattak, Abdul Ghafoor Sajjad, Syed Ali Hussain, Zouina Sarfraz, Azza Sarfraz, Ivan Cherrez-Ojeda

**Affiliations:** 1Department of Rehabilitation Sciences, Abasyn University, Islamabad 44000, Pakistan; 2Department of Rehabilitation Sciences, The University of Lahore, Islamabad 44000, Pakistan; 3Department of Rehabilitation Sciences, Shifa Tameer-e-Millat University, Islamabad 44000, Pakistan; 4Department of Research and Publications, Fatima Jinnah Medical University, Lahore 54000, Pakistan; 5Department of Pediatrics and Child Health, The Aga Khan University, Karachi 74000, Pakistan; 6Department of Allergy, Immunology & Pulmonary Medicine, Universidad Espíritu Santo, Samborondón 092301, Ecuador

**Keywords:** lumbar disc, surgery, rehabilitation, functional improvement, pain score, global improvement

## Abstract

**Introduction**: Lumbar disc surgery is a common procedure for patients with lower back pain associated with lumbar disc herniation. This study aims to evaluate the impact of active physiotherapeutic rehabilitation on global/functional improvement and subjective pain score reduction among patients 1–2 months following lumbar disc surgery. The outcomes of this study are to assess the impact of active physiotherapeutic rehabilitation on functional improvement and subjective improvement in pain behavior post active rehabilitation. The outcomes are measured as pain assessed using the visual analog scale, global measurement of improvement, back pain functional status, and return to work. **Methods:** Databases, including MEDLINE/PubMed (10 June 1996, 2022), Web of Science (10 June 1997, 2022), Scopus (15 March, 10 June 2004, 2022), CINAHL Plus (10 June 1961, 2022), and Cochrane (10 June 1993, 2022) were reviewed without any language restrictions. All studies were systematically screened; however, only randomized controlled trials were eligible against the inclusion/exclusion criteria. All statistical tests were conducted in Review Manager (RevMan) 5.4. The quality of studies was appraised using the grading of recommendations assessment, development, and evaluation (GRADE) approach and the risk-of-bias 2 (RoB 2) tool. **Results:** Fifteen articles were identified, enrolling a total of 2188 patients, where the majority of active rehabilitation interventions continued for 3 months. All these interventions began 1–2 months postoperatively, and quantitative findings were presented as mean scores. The subjective pain scores were significantly lower in the interventional group, with a mean difference (MD) of −7.01 (*p* = 0.004). The pain disability score was considerably lower in the interventional group, with an MD of −3.94 (*p* = 0.002). Global improvement was higher in the interventional group (OR = 1.94, *p* = 0.0001). **Conclusions**: This study presents significant improvement in all parameters concerning pain and functionality. Postoperative rehabilitation requires optimization concerning timing, duration, intensity, and associated components to benefit patients post lumbar disc surgery.

## 1. Introduction

Lumbar disk herniation (LDH) commonly manifests among young and middle-aged patients and is the most common reason for lumbar surgery [1]. Herniation refers to the displacement of intervertebral disk material beyond the normal margins of the disk position, which may be either a biochemical or mechanical process [1]. Whereas the incidence of LDH is likely to be underreported due to its potential asymptomatic nature, low-grade (level IV–V) evidence suggests that approximately 90% of patients resolve their symptoms without substantial interventions [2]. The literature favors both conservative management and surgical intervention for LDH [3]. Surgical interventions promise faster relief of symptoms and an earlier return to function, with long-term outcomes being similar irrespective of the type of management [4]. When considering surgical options, discectomy is the most commonly performed surgery for LDH, with strong evidence of its clinical effectiveness [5]. During the procedure, the portion of the disc that causes pressure on the nerve root is removed, and in some cases, we may remove the entire disc. However, only an estimated 70% of patients are fit to return within 12 months following lumbar surgery [6].

The World Health Organization (WHO) defines physical rehabilitation as “a set of interventions designed to optimize functioning and reduce disability in individuals with health conditions in interaction with their environment” [7]. Physical rehabilitation is an essential component of universal health coverage along with the treatment and promotion of good health [7]. At present, an estimated 2.4 billion across the globe are living with health conditions that could benefit from rehabilitation [7,8]. The need for physical rehabilitation worldwide is predicted to increase due to the change in the health characteristics of the population. For example, people tend to live longer, but with more chronic diseases and disabilities [7,9]. The rationale behind this systematic review and meta-analysis is to further explore the impact that ‘active’ physiotherapeutic rehabilitation (i.e., a recovery-based program, where the client plays an active role in increasing function and overall strength, which includes, but is not limited to, stability, mobility, strength, and endurance training through specific exercise and progression) has on pain and global/functional improvement 1–2 months after lumbar disc surgery.

Existing research has focused on various physical rehabilitation programs to improve the short-term outcomes following lumbar surgery [10]. A notable contribution in this area was published in a Cochrane review conducted by Oosterhuis and colleagues in 2014 [11]. The authors presented evidence for early physiotherapy rehabilitative measures that lead to improved treatment outcomes, initiating as early as 4–6 weeks post-surgery [11]. Typically, these physiotherapeutic rehabilitation programs involve specific exercise therapies led by physiotherapists to enable a quicker return to normal activities, such as walking or working, in post-surgical patients [12]. It is estimated that, while around 78–95% of patients tend to improve after surgery, approximately 3–12% of patients continue to have symptoms, and a subset of patients may require surgery again [11]. There is a paucity of systematic reviews/meta-analytical studies addressing whether patients warrant active physiotherapy rehabilitation after lumbar disc surgery in the current literature.

Therefore, this study aims to identify adult patients aged 18–65 and enlisted in active rehabilitation programs after having undergone first-time lumbar surgery with surgical techniques, including standard discectomy, laser discectomy, microdiscectomy, or chemonucleolysis. We hypothesize that there may be improved functionality and reduced subjective pain among patients undergoing active rehabilitation compared to standard care treatment at 1–2 months following the surgery. The primary outcome of this study is to evaluate the impact of active physiotherapeutic rehabilitation on functional improvement and the secondary outcome is to explore the subjective improvement in pain behavior after active rehabilitation.

## 2. Methods

### 2.1. Search Strategy

A detailed systematic search was conducted using key electronic databases following PRISMA 2020 statement guidelines. MEDLINE/PubMed (10 June 1996, 2022), Web of Science (10 June 1997, 2022), Scopus (15 March, 10 June 2004, 2022), CINAHL Plus (10 June 1961, 2022), and Cochrane (10 June 1993, 2022) were reviewed without language restrictions (any non-English study was to be translated to English using the Google Translate tool). The search terms across all databases comprised of a combination of the following, using BOOLEAN (and/or) logic. The following keywords were applied: Exercise Therapy *, Lumbar Vertebrae *, Diskectomy [methods, * rehabilitation], Intervertebral Disc [* surgery], Laminectomy [* rehabilitation], Postoperative Period, Randomized Controlled Trial, Recovery of Function. The titles and abstracts of shortlisted studies from the enlisted databases were screened independently by two reviewers. During the screening phase, the reference lists of the studies were assessed, applying the umbrella overview of studies to ensure that no data were omitted. In case of any disagreements, the third reviewer resolved them and enabled the team to reach a consensus. Cohen’s coefficient of the inter-reviewer agreement was calculated.

### 2.2. Types of Studies Included

Only randomized controlled trials were eligible for inclusion in this review. All other studies, including cohorts (retrospective or prospective), case series, case reports, systematic reviews, meta-analyses, letters, and brief reports, were excluded.

The inclusion criteria are as follows: adult patients aged 18–65 of any gender undergoing first-time lumbar disc surgery due to prolapse of the lumbar disc were included. Moreover, individuals undergoing any surgical technique, including standard or laser discectomy, microdiscectomy, and/or chemonucleolysis, were included.

The exclusion criteria were as follows: pediatric patients or those aged above 65 years not undergoing first-time post-surgical procedures, as elucidated above, were omitted.

The PICO framework is attached in Table 1 below.

### 2.3. Data Extraction (Selection and Coding)

The first two reviewers independently extracted data from the studies into a shared Google spreadsheet. A third reviewer was present for any disagreements. The pair identified the patients, interventions, outcomes of interest, and effect size of the screened studies. Once two reviewers independently scanned the studies, the third reviewer finally assessed the domains extracted from the spreadsheet and viewed the patient, intervention, primary, and secondary outcome measures. The reference lists of shortlisted studies were also assessed for eligibility for inclusion (umbrella methodology). As enlisted in Table 1, the primary outcome measures included: (i) pain changes using the visual analog scale, (ii) global measurement of improvement (overall improvement of health, subjective test to quantify improvement, proportion of sample size showing recovery), (iii) back pain functional status (Oswestry Disability Index, Roland Morris Disability Questionnaire), and (iv) return to work (days off work, return to work status).

The secondary outcome measures of the physical examination pertained to the spinal range of motion, muscle strength, and straight-leg raise range of motion; the behavioral outcomes included anxiety, depression, and pain behavior

The individual study data were entered into a presentable format during the inclusion phase, and the clinical relevance assessment was also conducted. The data software EndNote X9 (Clarivate, London, UK) was utilized to omit duplicates during the study selection process. The bibliography software utilized for this was Mendeley (Elsevier, Amsterdam, The Netherlands), where all included RCTs were recorded and organized. The Kappa score, an inter-rater reliability measure of agreement between independent raters, was also computed using Statistical Package for Social Sciences (SPSS, V. 24). The meta-analysis used Review Manager 5.4 to compute mean differences and odds ratios using 95% confidence intervals. The findings of the meta-analysis were presented as forest plots with *p*-values presented. The funnel plot was additionally generated to visually assess for publication bias. All statistical tests were conducted utilizing Review Manager (RevMan) 5.4 (Cochrane, London, UK).

### 2.4. Risk of Bias (Quality) Assessment

The included RCTs were assessed for homogeneity and heterogeneity of the study population, treatment, outcomes, and measurement instruments. If one outcome measure was heterogeneous, a narrative summary of findings was presented by critically appraising the differences. The GRADE approach as listed by Cochrane Training was used to assess the overall quality of evidence [13]. The factors that impacted the quality of evidence include the risk of bias, study design, inconsistent results, lack of generalizability, and inaccurate data. The quality of evidence was graded as: 1: high-quality evidence, 2. moderate quality evidence, 3. low-quality evidence, 4. very low-quality evidence, and 5. no evidence; the findings are listed in Table 2. The GRADE assessment form was shared with each author, and the final scores were agreed upon before synthesis.

Version 2 of the Cochrane risk-of-bias tool for randomized trials (RoB 2) was utilized to assess the risk of bias in all the included studies [14]. The RoB 2.0 assessment comprised five domains, as follows. (1) Bias arising from the randomization process; (2) Bias due to deviations from intended interventions; (3) Bias due to missing outcome data; (4) Bias in the measurement of the outcome; and (5) Bias in the selection of the reported result. Domain-level judgments about risk of bias were classified as the following: (1) Low risk of bias; (2) Some concerns; and (3) High risk of bias. The traffic light plot of bias assessment and the weighted summary plot of the overall type of bias encountered is illustrated in Section 3.3: risk of bias synthesis.

### 2.5. Protocol Registration and Role of Funding

This systematic review and meta-analysis protocol was registered with PROSPERO ID: CRD42021285371. No funding was obtained.

## 3. Results

Of the 1394 studies identified from databases, all were screened. Post-screening, 1258 studies were excluded, and 136 full-text studies were assessed for eligibility (Appendix A). Finally, we included 15 trials in this systematic review (Figure 1). Kappa’s score was calculated to be 0.93.

In this systematic review, we identified fifteen trials, including 2188 patients. Of these trials, only three had low-quality evidence, whereas five had high-quality evidence, and six were of moderate quality. Both males and females were included, with a representation of 1250 (57.1%) and 938 (42.9%) patients, respectively. All patients in this systematic review underwent standard discectomy and microdiscectomy (i.e., lumbar disc surgery). The majority of the rehabilitation interventions continued for 3 months. All interventions began postoperatively between 4–8 weeks. All studies that had the rehabilitation start immediately after surgery or 1 year post-surgery were omitted to ensure uniformity of findings. The characteristics of included studies are depicted in Table 2, along with the GRADE scores.

**Table 2 healthcare-10-01943-t002:** Characteristics of included studies.

Sr. No.	Author, Year	Aim	Methodology	Participants	Interventions	Outcomes	GRADE Scores
1	Atlanta et al., 1986 [15]	To examine the one-year postoperative results in patients operated on for lumbar disc herniation with comprehensive rehabilitation and normal care facilities.	Randomization by age (>40 years) and sex stratification before the operation; N = 212	Participants underwent first-time disc surgery for lumbar prolapse patients; the operation was carried out through an interlaminar trepanation and sequesters, and any loose nucleus pulposus material was removed.	IG = PR started four weeks after surgery (N = 106): multifactorial rehabilitation (physiatrist, physical and occupational therapists, psychologist, social worker) for two weeks, “Intensive Back School.” Encouraging physical activities; CG = Usual care.	At 1-year follow-up—Global perceived effect (five-point scale): ‘Much better or Better’-IG = 88%, CG = 83%. Occupational handicap (WHO scale) and total sick leave during a one-year follow-up period. No significant differences between groups. Reoperations: IG = 4/106, CG = 4/106.	Moderate-quality evidence
2	Donceel et al., 1999 [16]	To compare rehabilitation-oriented approaches focused on early mobilization and early resumption of professional activities.	Randomization was done by computer-generated number; N = 710	The mean age of participants was 39.2 years and patients underwent open lumbar discectomy. Rehabilitative interventions started six weeks post-surgery.	IG = At first visit six weeks after surgery, functional evaluation, natural history, and expected work incapacity was discussed. Patients were encouraged and stimulated with personal activities and early mobilization—CG = Usual care.	On return to work at the 52nd week post-intervention, improvement was noted in 89.9% (IG) and 81.9% (CG) of patients; the differences were significant.	High-quality evidence
3	Danielsen et al., 2000 [17]	To assess the effect of an early regimen of vigorous medical exercise compared with an ordinary care program.	Randomization by a number table; N = 63.	The patients were aged 22 to 58 years, four weeks after the operation for lumbar disc herniation (arcotomy in 36 patients, microsurgical in 27 patients.	IG = Enrolled into the rehabilitation program from weeks 4–12, three times per week × 40 min per session exercise therapy; exclusively active, no manual intervention or physical therapist, strengthen muscles (various apparatus), participant tailored; N = 39. CG = weeks 1–3 comprised of standard programs, follow-up consultation for clinical course and clinical examination with a physical therapist every two weeks for eight weeks, with a mild home exercise program, relaxing and resting the back, and gradually resuming daily activities; N = 24.	Pain intensity (VAS) at 6 months: IG = 3.7 (95% CI: 2.7–4.7), CG = 2.0 (95% CI: 0.7–3.3). Functional status (RDQ): IG = 8.9 (7–10.8), CG = 5.4 (3–7.8). Pain at 12 months: IG = 3.2 (2.1–4.3), CG = 1.8 (0.5–3.1); (RDQ) IG = 8.7 (6.8–10.6), CG = 5.3 (2.6–8). Absolute RDQ values had a minor advantage for IG at 6 and 12 months. Pain scores were significantly better for IG at 6 months. A larger no. of participants in IG resumed daily activities (WONCA scale) at 6 months.	High-quality evidence
4	Dolan et al., 2000 [18]	To determine the effects of a postoperative exercise program on pain, disability, psychological status, and spinal function.	Blind randomization; N = 20.	The patients were aged between 18 and 60 with radiological evidence of disc prolapse associated with sciatica of fewer than 12 months duration; they underwent microdiscectomy, followed by six weeks of routine postoperative care. In the exercise group, the IG underwent a 4-week exercise program concentrating on improving strength and endurance of the back and abdominal muscles and mobility of the spine and hips.	IG = Underwent an exercise program by an experienced physiotherapist, two one-hour sessions per week for four weeks (commenced six weeks after surgery); there were general aerobic exercises, stretching exercises, extension exercises, strength and endurance exercises (back and abdominal); N = 9. CG = acquired no further treatment; N = 11	Pain intensity (VAS): Reductions were noted in both CG and IG 6 weeks after surgery, but IG showed a further decrease. Between IG and CG, at 12 months, pain (diary): was significantly less in IG (*p* < 0.05). Functional status (ranging from 0–75) was improved in both groups: IG = 54 ± 24, CG = 50 ± 25. Little behavioral outcome changes in IG and CG, with no differences in muscle endurance and ROM.	Low-quality evidence
5	Ostelo et al., 2003 [19]	To note the effectiveness of behavioral graded activity provided by physiotherapists for patients after first-time lumbar disc surgery.	Randomization was done a-priori; N = 105.	The patients were aged 18 to 65, still suffering complaints six weeks post-surgery, which had restrictions in normal activities of daily living.	IG: Using graded activity and positive reinforcement with time contingency management, the patients underwent behavioral graded activity (operant therapy); N = 52. Using baseline measurements, the individual-graded exercise training program was tailored. There was a total of 18 × 30-min sessions over three months. CG = Usual care with exercise trunk muscles, mobilization exercises with 18 × 30 sessions; N = 53.	Global perceived effect: 65% in IG recovered compared to 62% in CG. Functional status (24-item RDQ): mean (SD); −6.4 (5.8) ~IG vs. −6.1 (5.6) ~CG. Pain back (VAS): −13.7 (31.4) vs. −20.9 (31.6).	Moderate-quality evidence
6	Yilmaz et al., 2003 [20]	To determine the efficacy of dynamic lumbar stabilization exercises in patients with lumbar microdiscectomy.	Patients were randomized into three groups; N = 42.	The patients were between 20 and 60 years old, including one month after first-time lumbar disc surgery. The follow-up was short-term.	IG-1 = Underwent dynamic lumbar stabilization exercise for eight weeks under supervision; N = 14. IG-2 = Partook in the flexion–extension (Williams–McKenzie) home program for eight weeks; N = 14. CG = No treatment.	Pain (VAS scores): IG-1 = 1.14 ± 0.86, IG-2 = 2.93 ± 2.02, CG = 4.29 ± 1.9. Functional status (scores on Modified Oswestry at post-treatment): IG-1 = 8.5 ± 4.8, IG-2 = 12.93 ± 4.23, CG = 17.71 ± 6.23.	Low-quality evidence
7	Choi et al., 2005 [21]	To identify the effects of postoperative early isolated lumbar extension muscle-strengthening program on pain, disability, return to work, and back muscle power after operation for the herniated lumbar disc.	Patients were randomized; N = 75.	The mean age of the patients was 46.09 years; they had leg pain not responding to conservative treatment, and had undergone discectomy.	IG = Advice, lumbar extension handout, home exercise for six weeks, and intensive training for 12 weeks (N = 35). MedX system, which restricts hip and pelvic motion. CG = Advice, lumbar extension handout, six weeks of home-based exercise, continued for another 12 weeks (N = 40).	Pain intensity (VAS) largely decreased in both IG (2.51) and CG (4.3) after 12 weeks of extension exercise (*p* < 0.05). Functional status (ODI): Postoperative ODI scores were improved compared with preoperative ODI scores IG = 24.6 and CG = 30.6 post-treatment (non-significant). Return to work: More than 92% returned to work within six months after surgery; within four months, 87% from IG and 24% from CG returned to work.	Moderate-quality evidence
8	Filiz et al., 2005 [22]	To compare two different exercise programs versus a control group, after lumbar disc surgery.	Randomized into three groups based on a blinded envelop-based system; N = 60.	Across all three arms, patients were included one month after first-time lumbar disc surgery, aged between 20 and 50 years. The follow-up was short-term in nature.	IG-1 = Underwent an intensive exercise program and back school education under supervision for eight weeks, three days a week with sessions of 1.5 h each, N = 20; IG-2 = were given back education and trained on McKenzie and Williams exercise with a home program for eight weeks to practice, N = 20; CG = No treatment, N = 20.	Pain (post-treatment score on VAS): IG-1 = 4.5 ± 1.6, IG-2 = 12 ± 3.7, CG = 13.3 ± 7.3. Functional status (post-treatment scores on Modified Oswestry): IG-1 = 7.1 ± 4.9, IG-2 = 11.7, CG = 15.1 ± 8.6. RTW (in days): IG-1 = 56.07 ± 18.66, IG-2 = 75 ± 24.9, CG = 86.2 ± 27.1.	Moderate-quality evidence
9	Hakkinen et al., 2005 [23]	To assess the adherence to and effects of a 12-month combined strength and stretching home exercise regimen versus stretching alone on patient outcome after lumbar disc surgery.	Patients were randomly assigned to IG and CG; N = 126.	The patients were enrolled two months after their first lumbar disc surgery; they were not pain-free (VAS > 10 mm).	IG = Placed with a home-based exercise program for 12 months. The patients were instructed to stretch and stabilize three times, with strength training, instructed to perform two sets of exercises twice a week. CG = Was instructed on regular stretching and stabilization three times.	At 12 months, improvement in back pain (100-mm VAS) was noted in IG = 4 mm (−11 to 5), CG = 1 mm (−7 to 9), leg pain (100-mm VAS), IG = −2 (−7 to 7), CG = −2 (−7 to 3). Improvement in disability (ODI) was IG = 3 mm (−6 to 1), CG = −2 (−5 to 1).	High-quality evidence
10	Donaldson et al., 2006 [24]	To compare the outcomes of formal post-surgical exercise-based rehabilitation to usual post-procedural surgical advice.	Randomization was carried out into two groups; N = 93.	The patients had a mean age of 41 and had standard open lumbar discectomy via the Spengler technique.	IG = Intervened at six weeks post-op and underwent 6-months of progressive training with 3sets of repetitions per exercise. There were 3 phases: conditioning, hypertrophy, and strength; N = 47. CG = Was given surgical advice; N = 46.	All values were noted at 58 weeks. Functional status (ODI): IG = 11.66 ± 2.25, CG = 12 ± 1.84. Functional status (RMDQ): IG = 4.03 ± 0.91, CG = 4.53 ± 0.74. The differences in SF36 [physical and mental category] were non-significant. Median time to return to work IG = 35 days and CG = 37 days.	Moderate-quality evidence
11	Erdogmus et al., 2007 [25]	To test the effectiveness of physiotherapy-based rehabilitation after lumbar disc surgery.	Randomized into three groups; N = 120.	The patients had a mean age of 41.3 years; they underwent standard laminectomy and micro-discectomy.	IG-1 = Underwent physiotherapy-based rehabilitative program starting 4–6 weeks postoperatively for a total of 12 weeks; N = 40. IG-2 = Underwent sham therapy ‘neck massage’ for 30 min per session; N = 40. CG = standard care; N = 40.	Post-treatment scores on functional status (LBPRS): IG-1 = −15.98 (−18.02 to −13.9), IG-2 = −13.23 (−15.35 to −11.1), CG = −12.15 (−14.59 to −9.71).	High-quality evidence
12	Kulig et al., 2009 [26]	To examine the effectiveness of a new interventional protocol to improve functional performance in patients who have undergone a single-level lumbar microdiscectomy.	Patients were randomly allocated to receive education only or exercise and education; N = 98.	The participants had a mean age of 40.3 years and underwent microdiscectomy.	IG = Enrolled into the USC Spine Exercise Program and a ‘back care education session. The intervention started 4–6 weeks after surgery, with three months of training sessions per week. The sessions involved back extensor strength and endurance training (using a variable-angle Roman chair) and mat and therapeutic exercise training; N = 51. CG = had a one-hour back care education single session; N = 47.	Functional status (ODI): IG = −18.4 (−22.5 to −14.3), CG = −9.4 (−13.0 to −5.8).	Moderate-quality evidence
13	McGregor et al., 2011 [27]	To evaluate the benefits of rehabilitation and an education booklet in the postoperative management of patients undergoing discectomy or lateral nerve root decompression, compared to standard of care.	The patients were randomized into four groups; N = 338.	The overall mean age of all participants was 53.75 years. The patients underwent routine discectomy to confirm root symptoms using signs and MRI results of lumbar disc herniation.	IG-1 = Intervention started at 6–8 weeks post-op, consisting of twelve, 1-h classes including aerobic fitness, stability exercises, stretching, stability exercises, endurance and strengthening training for the back, abdominal, and leg muscles, ergonomic training, and advice on living and setting targets; N = 86. IG-2 = Underwent rehabilitation and booklets; N = 91: IG-3 = Were intervened with educational booklets ‘Your Back Operation’; N = 70. CG = routine practice; N = 91.	Functional status (12-month scores on ODI). IG-1 = 24 ± 21, IG-2 = 26 ± 22, IG-3 = 25 ± 20, CG = 27 ± 23. Pain intensity (VAS). IG-1 = 72 ± 24, IG-2 = 71 ± 26, IG-3 = 72 ± 23, CG = 67 ± 26.	High-quality evidence
14	Oestergaard et al., 2013 [28]	To examine the effect of early initiation of rehabilitation after instrumented lumbar spinal fusion.	Block randomized; two groups intervened at 6 and 12 weeks post-surgery with no control group; N = 82.	The mean age of all participants was 52 years; all patients planned for an instrumented lumbar spinal fusion due to degenerative disc disease or spondylolisthesis grade I or II were randomized at 6 or 12 weeks after surgery.	IG = Enrolled at 6 or 12 weeks post-surgery, with each session targeting pain and physical incapacity, problems, and solutions in performing ADL. The physiotherapist advised home exercises, focusing on active stability training of the truncus and large muscle groups.	Functional status (ODI) at 6-month follow-up. IG-1, the 6-week group, had achieved a reduction of 6 points; IG-2, the 12-week group, achieved a decrease of 15 points. At 1-year follow-up, IG-1 had a decrease of 5 points, and IG-2 had a reduction of 20 points.	Low-quality evidence
15	Demir et al., 2014 [29]	To investigate the effects of supervised dynamic lumbar stabilization exercises during postoperative rehabilitation on pain, spinal mobility, and functional status among patients undergoing lumbar microdiscectomy for the first time.	Randomization into two groups; N = 44	The mean age of all patients was 41.1 years. They were randomly divided into two equal groups (N = 22), with the IG focusing on dynamic lumbar stabilization and home exercises. CG underwent home exercises only for four weeks in total.	IG = Partook in home exercises and DLA at the 4th week post-operation and the activities continued for four weeks. CG = Was part of the home exercise program comprising stretching, pelvic tilt, extension, and flexion strengthening of the trunk and abdomen.	Functional status (ODI) at 1-month follow-up. IG = 17.5 (IQR = 15.2), CG = 23 (IQR = 11.5). Back Pain (VAS) in 1st month. IG = 10 (IQR = 20), CG = 10 (IQR = 20)	Moderate-quality evidence

CG: Control group; IG: Interventional group; PR: Physical rehabilitation; WHO: World Health Organization.

### 3.1. Meta-Analytical Findings

#### 3.1.1. Back Pain Functional Status Post-Treatment

A total of 8 of the 15 trials documented data for functional status of back pain post-treatment, comprising 302 patients in the intervention group and 247 in the control group (N = 549). A negative value was computed on noting the mean difference (MD = −3.94, 95% CI = −6.53, −1.35; *p* = 0.002), meaning that the pain disability score was lower in that intervened physiotherapy rehabilitation (Figure 2A). The outcome was assessed for effect size with values reported as Cohen’s d. The effect size was medium in favor of the intervention group showing a reduction in back pain with improved functional status post-treatment (Cohen’s d = −0.57, 95% CI = −0.92, −0.22; *p* = 0.002) (Figure 2B).

#### 3.1.2. Global Measure of Improvement

A total of 6 of the 15 studies reported data on a ‘global measure of improvement’ where the proportion of patients that showed improvement was documented. In total, 617 patients in the intervention group and 628 patients in the control group were analyzed (N = 1245). The odds ratio (OR) was computed, where it was noted that the intervention group had a higher likelihood of improvement (OR = 1.94, 95% CI = 1.38, 2.72; *p* = 0.0001). Very low heterogeneity was noted based on the I^2^ test result (I^2^ = 11%) (Figure 3).

#### 3.1.3. Pain Scores Post-Treatment

A total of 9 of the 15 studies reported mean scores post-treatment, comprising a total of 325 patients in the intervention group and 283 in the control group (N = 608). The mean difference (MD) is a standard statistic that measures the absolute differences between the intervention and control groups’ pain scores. On noting the mean difference (IV, Random, 95% CI), a negative difference was found (MD = −7.01, 95% CI = −11.84, −2.18; *p* = 0.004), meaning that the pain score was significantly lower in the intervention group (Figure 4A). A large effect size was computed in favor of intervention for pain scores post-treatment (Cohen’s d = −0.89, 95% CI = −1.49, −0.29; *p* = 0.004) (Figure 4B).

### 3.2. Funnel Plot

As noted in Table 2, 5 of 15 trials had high-quality evidence applying the GRADE scoring system. 6 of 15 trials were noted to have moderate-quality evidence, with three trials falling at low-quality evidence. The overall quality of the studies employed in this systematic review is in the moderate-quality range. A funnel plot is depicted in Figure 5 to assess for publication bias. It may be seen that around half of the studies deviate from an inverted funnel shape, meaning that there is a risk of some studies being underrepresented in the literature.

### 3.3. Risk of Bias Synthesis

On noting the bias arising from the randomization process, 12 studies had low concerns, two had some concerns, and one had a high concern. On judging biases due to deviations from the intended interventions, eight had low risk, whereas seven had some concerns. When assessing bias due to missing outcome data, seven had some concerns, whereas six studies were with low concerns and two with high concerns. When noting bias in the measurement of the outcome, nine studies had low concerns, whereas six had some concerns. For bias in the selection of the reported result, 10 had low concerns, whereas five had some concerns. Overall, eight studies had some concerns, five had low concerns, and two had high concerns (Figure 6).

## 4. Discussion

We included 15 RCTs in our systematic review and meta-analysis. The studies included in our synthesis were similar in that the active physiotherapeutic rehabilitation programs were offered between 1–2 months postoperatively. Overall, our findings suggest that programs within the specified period show moderate improvement in functional recovery and subjective pain compared to no interventions. These improvements were noted among the interventional group for the subjective pain scores, global improvement measures, and functional status related to back pain. The trials we included were of moderate-to-high-quality evidence, with a mild–moderate publication bias. The heterogeneity between the studies can be attributed to the differences in type, duration, and timing of the interventions administered. We are uncertain of the nature of postoperative care patients in the control group across the studies. Protocols focusing on at-home interventions by physical therapists are already being offered to patients, which may lead to a negative skewing of effect sizes against the interventional group [30]. Therefore, we provide moderate–high-quality evidence of active physiotherapeutic rehabilitation programs to recovery started 1–2 months postoperatively [11].

Postoperative physiotherapeutic rehabilitation not only consists of multidisciplinary exercise programs with group and individual sessions, but also affixes patient education and psychosocial interventions. In the trials we analyzed, frameworks for the interventions consisted of cardiovascular exercise, motor control/stability, strength training, joint mobilization/stretching, and nerve/soft-tissue mobilization. The trials employed a multidisciplinary approach that promoted active rehabilitation following lumbar disc surgery. None of the trials reported any specific adverse events that warranted stopping rehabilitative programs, e.g., increased complication rates. The rationale for early and active rehabilitation is to accelerate return to work among patients. Our findings combine heterogeneous evidence regarding duration, intensity, type of intervention, and assessment time.

Implementing active physiotherapeutic rehabilitation after lumbar disc surgery has shown positive effects on patient recovery. Patients have typically had high avoidance of physical activities following lumbar disc surgery, which has been shown to reduce their quality of life [31]. For instance, a study reported approximately 50% of the patients had kinesiophobia 10–34 months following lumbar disc surgery [32]. Muscle function is often concurrently compromised, which is not corrected after lumbar disc surgery, suggesting a strong role of exercise in the early postoperative period that ties in with active and early mobilization [33]. Restoration of function may be possible with physical activity to desensitize and normalize sympathetic feedback in the affected region [34]. Therefore, it is important to encourage early mobilization to avoid potential complications and muscular atrophy following lumbar disc surgery. Additionally, these may suggest the importance of a biopsychosocial approach for recovery after surgery and consistently encouraging patients to resume activities within 1–2 months.

### 4.1. Current Evidence and Key Underpinnings

In this study, we discussed the role of active rehabilitation. However, it ought to be noted that other options ought to be considered among the spectrum of care for patients with lumbar surgery. At present, evidence from systematic reviews has examined that active rehabilitation for lumbar stenosis post decompression surgery has been more effective than standard care, both in the short term and the long term, concerning functional status [35,36]. There is currently limited agreement by societies worldwide concerning physical therapy, the timing, and the physiological mechanism associated. For instance, the wide variety of interventions tends to differ in terms of delivery (intensity and duration) [12,37]. Another example may be taken from a systematic review that posits that starting physical therapy during the 12-week postoperative period leads to better outcomes, along with lower costs, compared to starting in the sixth week [31]. Nonetheless, the review does not consider the role of active physiotherapeutic rehabilitation, which is why the findings of the “active component” become imperative. It ought to also be reiterated that psychosocial support also improves all listed outcomes. While the data we have collated served as the best evidence available when considering active rehabilitation within 1–2 months, a shortcoming of this meta-analysis is that we could not segregate studies as the first month versus the second month as the commencement data due to the misalignment of the data with our aims/objectives.

### 4.2. Clinical Significance

The findings of this systematic review and meta-analysis are a guiding force for future clinicians wishing to explore the timelines upon which active physiotherapy rehabilitation may be commenced to improve pain and global and functional improvement post lumbar disc surgery [38]. Future studies may utilize the various outcome measures in our meta-analysis to quantify available outcomes and to improve key knowledge gaps in the current literature to advance the understanding of the term “active” rehabilitation [39,40]. The “active” component of physiotherapeutic rehabilitation is less understood by patients owing to a paucity of concrete societal guidelines on this intervention. In simple terms, active rehabilitation focuses on playing a non-static role in resuming function and strength; this tends to improve stability, strength, endurance, and exercise progression [41]. Our meta-analysis summarizes quantitative evidence available for physiotherapists, surgeons, researchers, and patients.

### 4.3. Limitations

Considering the baseline differences and variable approaches taken for rehabilitation, we found mild-to-moderate improvement for patients initiated with active physiotherapeutic rehabilitation 1–2 months postoperatively. There are a few limitations of the included data that must be noted. The majority of the data were self-reporting. We aimed to categorize improvement in physical ranges, namely spinal range of motion, muscle strength, straight-leg raise range of motion, and behavioral outcomes, including anxiety and depression. However, none of the trials reported physical and behavioral effects. Having a more objective measure of functionality, such as physical improvement, may offer more insight into improvement with physical rehabilitation. The trials primarily report the mean differences in pain and function, which is beneficial, but further research ought to also report absolute improvement. It would be beneficial to indicate the baseline scores and progress at different follow-up time points to prevent misinterpretation. The current data do not provide specific subgroup analyses based on baseline health, psychosocial considerations, and degree of invasiveness of the surgery. It is possible to provide cost-effective postoperative rehabilitation to particular subgroups of patients who are most likely to benefit from more insight.

## 5. Conclusions

In conclusion, we collated evidence on the benefits of active physiotherapeutic rehabilitation for improvement in pain and functionality among patients following lumbar disc surgery. Having pooled in 2188 patients, where the majority of them commenced active physiotherapeutic rehabilitation within the 1–2-month period following lumbar disc surgery, we found quantifiable evidence that the pain scores were significantly lower with active intervention (*p* = 0.004). Furthermore, the pain disability score reduced (*p* = 0.002) and the global improvement was critically high (*p* = 0.0001). Overall, there are conflicting views on the rightful commencement timing for physiotherapeutic measures; however, we find that active physiotherapeutic rehabilitation optimizes pain and functionality within the 1–2-month period. However, our findings did not differentiate the baseline characteristics of patients, including psychosocial factors, health status, and invasiveness of the surgery. We considered the overall evidence to be of good quality considering the heterogeneity across the trials. We identified differences in the rehabilitation protocols across studies. Therefore, we recommend a combined effort to optimize timing, duration, intensity, and associated components of rehabilitation among patients who underwent lumbar disc surgery to promote optimal recovery.

## Figures and Tables

**Figure 1 healthcare-10-01943-f001:**
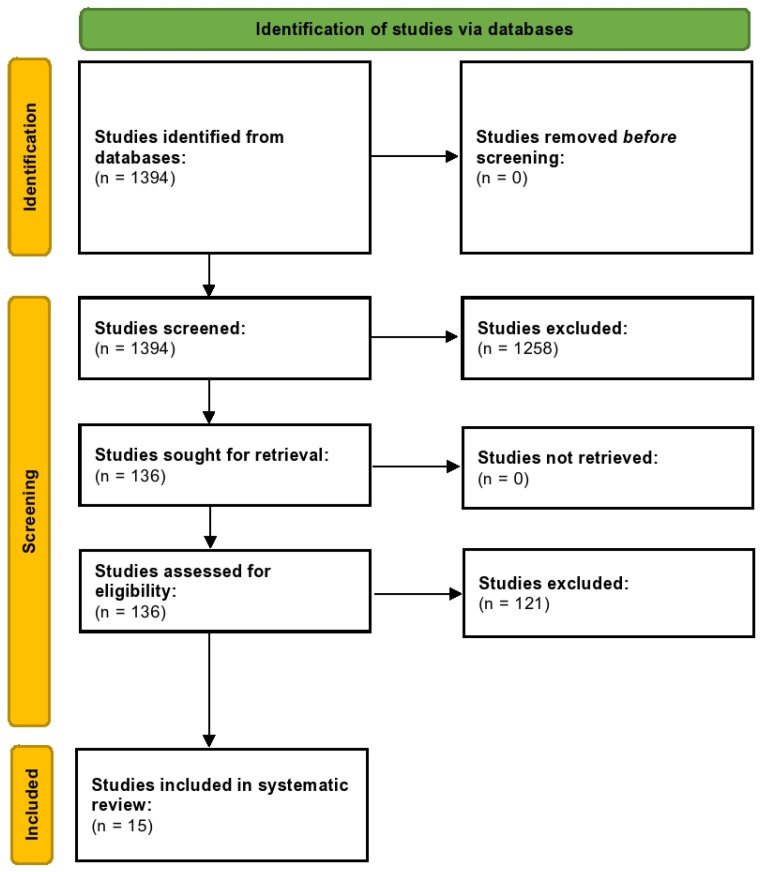
PRISMA flowchart.

**Figure 2 healthcare-10-01943-f002:**
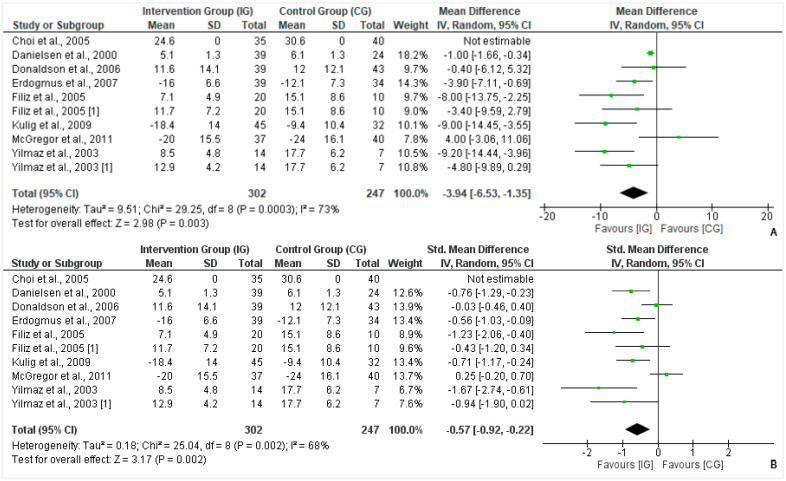
(**A**): Forest plot for functional status post-treatment (mean values [SD] IG versus CG). Heterogeneity: Tau^2^ = 9.51; Chi^2^ = 29.25, df = 8 (*p* = 0.0003); I^2^ = 73%. Test for overall effect: Z = 2.98 (*p* = 0.003) [17,20,21,22,24,25,26,27] (**B**): Forest plot for functional status post-treatment (Standardized mean difference [SMD] IG versus CG). Heterogeneity: Tau^2^ = 0.18; Chi^2^ = 25.04, df = 8 (*p* = 0.002); I^2^ = 68%. Test for overall effect: Z = 3.17 (*p* = 0.002) [17,20,21,22,24,25,26,27].

**Figure 3 healthcare-10-01943-f003:**
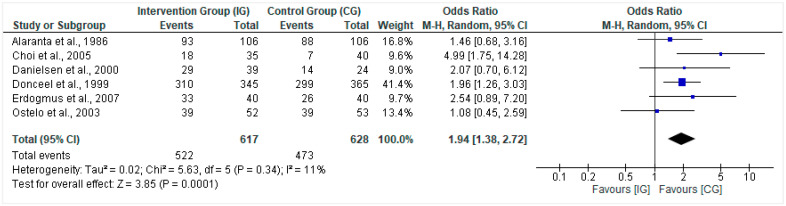
Forest plot for the proportion of patients with improvement (dichotomous data IG vs. CG). Heterogeneity: Tau^2^ = 0.02; Chi^2^ = 5.63, df = 5 (*p* = 0.34); I^2^ = 11%. Test for overall effect: Z = 3.85 (*p* = 0.0001) [15,16,17,19,21,25].

**Figure 4 healthcare-10-01943-f004:**
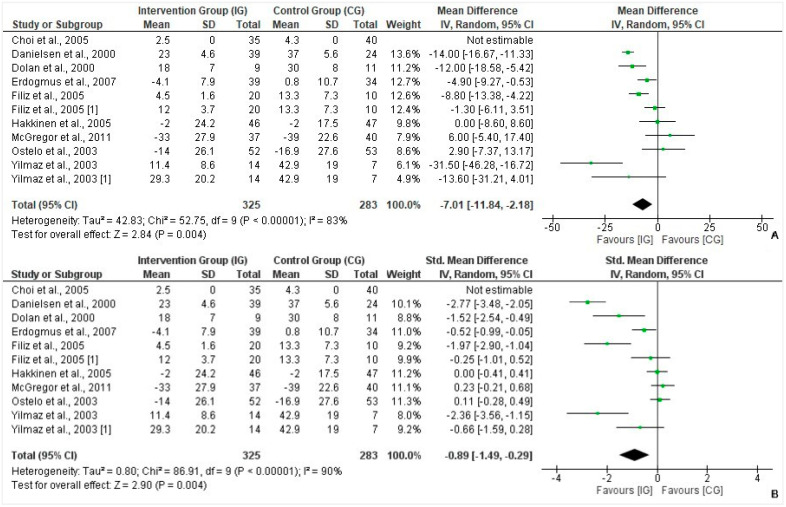
(**A**): Forest plot of pain scores (mean values [SD] IG versus CG). Heterogeneity: Tau^2^ = 42.83; Chi^2^ = 52.75, df = 9 (*p* < 0.00001); I^2^ = 83%. Test for overall effect: Z = 2.84 (*p* = 0.004) [17,18,19,20,21,22,23,25,27]. (**B**): Forest plot for functional status post-treatment (Standardized mean difference [SMD] IG versus CG). Heterogeneity: Tau^2^ = 0.80; Chi^2^ = 86.91, df = 9 (*p* < 0.00001); I^2^ = 90%. Test for overall effect: Z = 2.90 (*p* = 0.004) [17,18,19,20,21,22,23,25,27].

**Figure 5 healthcare-10-01943-f005:**
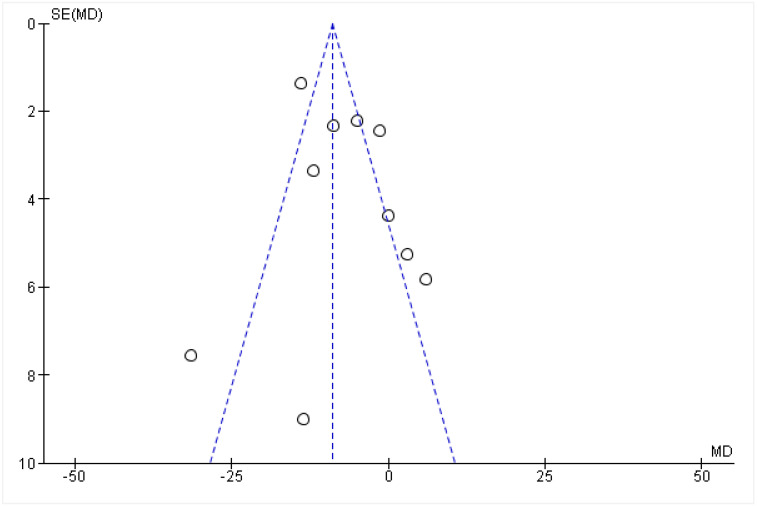
Funnel plot to visually assess for publication bias.

**Figure 6 healthcare-10-01943-f006:**
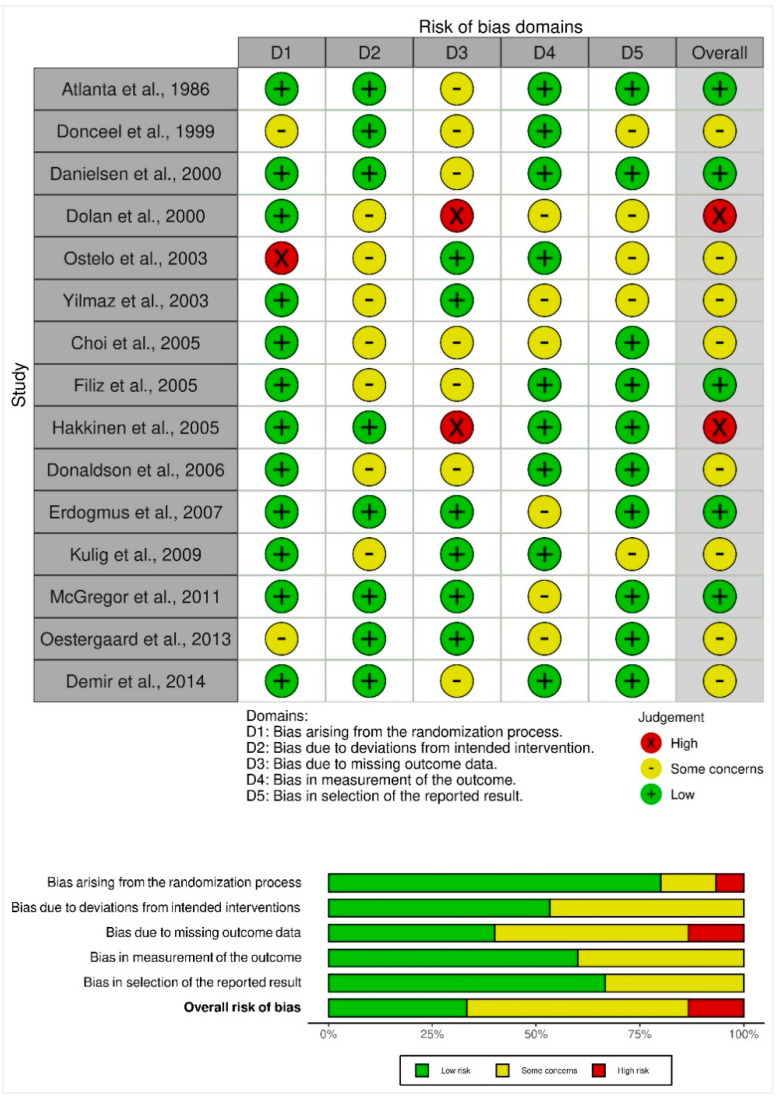
Risk of bias assessment of RCTs using the ROB-2 tool. Traffic light plot of study-by-study bias assessment. Weighted summary plot of the overall type of bias encountered in all studies [15,16,17,18,19,20,21,22,23,24,25,26,27,28,29].

**Table 1 healthcare-10-01943-t001:** PICO framework.

Participants	Intervention	Comparator	Outcomes
Adult patients aged 18–65 that were undergoing first-time lumbar disc surgery due to prolapse of the lumbar disc were included. Any type of surgical technique, whether standard discectomy, laser discectomy, microdiscectomy, or chemonucleolysis, was included. Studies that pertained to non-first-time post-surgical patients or those aged < 18 years or > 65 years were excluded.	Post lumbar disc surgery *active* rehabilitation programs include strength and mobility training, exercise therapy, physiotherapy, and multidisciplinary treatment. These programs may be conducted one-to-one or in a group-based setting. Individuals who do not undergo active enlisted programs and do not acquire physiotherapeutic measures are excluded.	The active physiotherapeutic rehabilitation was compared to a control group that did not acquire active rehabilitative treatment.	The randomized controlled trials were required to pertain to at least one of the four primary outcome measures, including: 1. Pain (i.e., visual analog scale), 2. A global measure of improvement (i.e., overall improvement of health, proportion of sample size showing recovery, subjective test to quantify improvement), 3. Back pain functional status (i.e., Oswestry Disability Index, Roland Morris Disability Questionnaire), and 4. Return to work (i.e., days off work, return to work status). The secondary outcomes of the physical examination pertained to the spinal range of motion, muscle strength, and straight-leg raise range of motion; the behavioral outcomes include anxiety, depression, and pain behavior.

## Data Availability

All data utilized for the purpose of this study are available publicly and online.

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
