# Peer review of "Impact of Active Physiotherapy Rehabilitation on Pain and Global and Functional Improvement 1–2 Months after Lumbar Disk Surgery: A Systematic Review and Meta-Analysis"

_healthcare, 2022, doi:10.3390/healthcare10101943_

Round 1

Reviewer 1 Report

Thank you for giving me this opportunity to review this article. The article is well written, though I have some serious concerns regarding the article.

Abstract:

1.               Mention the acronym of abbreviations when it is used for first time.

2.               Mention the databases searched for the review.

3.               Mention the search duration.

4.               Mention the character of the study articles.

5.               Mention the outcome measures measured for the review.

6.               Mention the results with a mean difference and odd’s ratio for each intervention procedure.

7.               The conclusion should be more concise and should not include abbreviations.

Manuscript

1.               Mention the application procedure, merits, and demerits of active rehabilitation therapy with recent references.

2.               The research question or the gap was not formulated properly with suitable recent references.

3.               Mention the clinical significance of this review to clinicians, researchers, and patients.

4.               The database searched for the review is much limited.

5.               Mention the keywords used for the search.

6.               The selection criteria are not specific – including exclusion criteria.

7.               Include the outcome measures analyzed for the review.

8.               Mention the data extraction procedure in detail.

9.               Mention the software used for the statistical analysis.

10.            Mention the kappa score of the reviewers who extracted the data.

11.            Include a table consisting of the number of articles extracted from each database.

12.            Include the mechanism (physiological) mechanism behind the changes in pain after physical therapy with recent references.

13.            Make the conclusion as per the data drawn from the reports.

14.            Mention the future recommendations of the study.

Author Response

Reviewer 1:

Thank you for giving me this opportunity to review this article. The article is well written, though I have some serious concerns regarding the article. 

Re. Abstract (Reviewer and Author Response Below):

Reviewer 1 Comment 1: Mention the acronym of abbreviations when it is used for the first time.

Author Response: Thank you very much for your valuable suggestion. I have made changes as suggested..

Reviewer 1 Comment 2: Mention the databases searched for the review.

Author Response: Thank you very much for your valuable suggestion. I have made changes as suggested.

“Databases including MEDLINE/PubMed (1996-June 10, 2022), Web of Science (1997-June 10, 2022), Scopus (March 15, 2004-June 10, 2022), CINAHL Plus (1961-June 10, 2022), and Cochrane (1993-June 10, 2022) were reviewed), without any language restrictions.”

Reviewer 1 Comment 3:  Mention the search duration.

Author Response: Thank you very much for your valuable suggestion. I have made changes as suggested.

“Databases including MEDLINE/PubMed (1996-June 10, 2022), Web of Science (1997-June 10, 2022), Scopus (March 15, 2004-June 10, 2022), CINAHL Plus (1961-June 10, 2022), and Cochrane (1993-June 10, 2022) were reviewed), without any language restrictions.”

Reviewer 1 Comment 4:  Mention the character of the study articles.

Author Response: Thank you very much for your valuable suggestion. I have made changes as suggested.

Reviewer 1 Comment 5. Mention the outcome measures measured for the review.

Author Response: Thank you very much for your valuable suggestion. I have made changes as suggested.

Reviewer 1 Comment 6: Mention the results with a mean difference and odd’s ratio for each intervention procedure.

Author Response: Respected reviewer, this point is invalid. For continuous outcomes, ONLY then can we report mean differences. For dichotomous outcomes, we can ONLY report Odd’s ratio (OR). The suggestion is statistically incorrect, hence no changes have been made. Please refer to the cochrane handbook for any confusion in this regard. 

Reviewer 1 Comment 7: The conclusion should be more concise and should not include abbreviations.

Author Response: It has been made more concise and any abbreviations if present were removed. Thank you very much for your valuable feedback.

Re. Manuscript (Reviewer and Author Response Below):

Reviewer 1 Comment 8:  Mention the application procedure, merits, and demerits of active rehabilitation therapy with recent references.

Author Response: Please refer to lines 62-67 and lines 53-55. Thank you for your feedback.

Reviewer 1 Comment 9: The research question or the gap was not formulated properly with suitable recent references.

Author Response: All recent references have been added.

Reviewer 1 Comment 10.  Mention the clinical significance of this review to clinicians, researchers, and patients.

Author Response: Thank you for your comment. This has been added.

Reviewer 1 Comment 11: The database searched for the review is much limited.

Author Response: This may be deemed invalid as PRISMA STATEMENT GUIDELINES 2020 require a minimum of 2 databases for a systematic review/meta-analysis to be considered valid. We have assessed 5 and also did an umbrella search. We are 100% confident that this is both a valid and reliable approach.

We have searched the following:

  1. PubMed/Medline
  2. Web of Science
  3. Scopus
  4. CINAHL Plus
  5. Cochrane

The databases are indeed fulfilling both our study’s aims and are aligned with current guidelines.

Reviewer 1 Comment 12:  Mention the keywords used for the search.

Author Response: Please review the highlighted text in the attached manuscript:

The following keywords were applied: Exercise Therapy*, Lumbar Vertebrae*, Diskectomy [methods, *rehabilitation], Intervertebral Disc [*surgery], Laminectomy [*rehabilitation], Postoperative Period, Randomized Controlled Trial, Recovery of Function.

Reviewer 1 Comment 13: The selection criteria are not specific – including exclusion criteria.

Author Response: We have defined the age group of patients, their gender, the first-time prolapse surgery AND type of surgeries. Please be kind enough to review:

The inclusion criteria are as follows: adult patients aged 18-65 of any gender undergoing first-time lumbar disc surgery due to prolapse of the lumbar disc was in-cluded. Moreover, individuals undergoing any surgical technique including standard or laser discectomy, microdiscectomy, and/or chemonucleolysis were included.

The exclusion criteria were as follows: pediatric patients or those aged above 65 years not undergoing first-time post-surgical -procedures as elucidated above were omitted.

Reviewer 1 Comment 14: Include the outcome measures analyzed for the review.

Author Response: Please review the highlighted text. It has been typed out for clarity.

As enlisted in Table 1, the primary outcome measures include i) pain changes using the visual analog scale, ii) global measurement of improvement (overall improvement of health, subjective test to quantify improvement, proportion of sample size showing recovery), iii) back pain-functional status (Oswestry Disability Index, Roland Morris Disability Questionnaire), and iv) return to work (days off work, return to work sta-tus).

The secondary outcome measures of the physical examination pertained to the spinal range of motion, muscle strength, and straight-leg raise range of motion; the behavioral outcomes include anxiety, depression, and pain behavior

Reviewer 1 Comment 15: Mention the data extraction procedure in detail.

Author Response: All details have been added. Thank you for your suggestion.

Reviewer 1 Comment 16:  Mention the software used for the statistical analysis.

Author Response: Has been added.

“All statistical tests were conducted utilizing Review Manager (RevMan) 5.4 (Cochrane).”

Reviewer 1 Comment 17: Mention the kappa score of the reviewers who extracted the data.

Author Response: Please review the first paragraph of the results. It is 0.93 and has been mentioned.

Reviewer 1 Comment 18: Include a table consisting of the number of articles extracted from each database.

Author response: Thank you for your suggestion. 

A supplementary table 1 has been added for this purpose.

Supplementary Table 1. Studies obtained from every database

PubMed/MEDLINE

Web of Science

Scopus

CINAHL Plus

Cochrane

278

442

489

158

27

Reviewer 1 Comment 19: Include the mechanism (physiological) mechanism behind the changes in pain after physical therapy with recent references.

Author Response: Thank you for your comment. This has been added.

Reviewer 1 Comment 20:  Make the conclusion as per the data drawn from the reports.

Author Response: Thank you for your comment. This has been added.

 Reviewer 1 Comment 21: Mention the future recommendations of the study.

Author Response: Thank you for your comment. This has been added.

Additional note for the respected reviewer 1: The paper has been read by all authors for typos/errors and an an ENL writer friend for final proofing. It should clear any worries for english quality.

Thank you for your hard work and dedication in readiny my work!

Regards

Reviewer 2 Report

The paper is well structured. I would suggest to make a difference between two groups: patients with physical therapy starteing after one month and two months after surgery in order to be able to recommend a timing to the surgeons. 

Author Response

Reviewer 2:

The paper is well structured. I would suggest making a difference between two groups: patients with physical therapy starting after one month and two months after surgery in order to be able to recommend a timing to the surgeons. 

Author response: While your comment is greatly appreciated, the data we have collated does not and could not be segregated as per your suggestion. There are far too many different weeks, etc, as the first month versus the second month concerning the commencement data of rehabilitation. Furthermore, this misaligned with our aims/objectives.

Additional note for the respected reviewer 2: The paper has been read by all authors for typos/errors and an an ENL writer friend for final proofing. It should clear any worries for english quality.

Thank you for your hard work and dedication in readiny my work!

Regards